# Smoke-Isolated Karrikins Stimulated Tanshinones Biosynthesis in *Salvia miltiorrhiza* through Endogenous Nitric Oxide and Jasmonic Acid

**DOI:** 10.3390/molecules24071229

**Published:** 2019-03-29

**Authors:** Jie Zhou, Zi-xin Xu, Hui Sun, Lan-ping Guo

**Affiliations:** 1School of Biological Science and Technology, University of Jinan, Jinan 250022, China; zhoujie8761@163.com (J.Z.); aaxzx1231@163.com (Z.-x.X.); sh15854141686@163.com (H.S.); 2State Key Laboratory of Dao-di Herbs, National Resource Center for Chinese Materia Medica, China Academy of Chinese Medical Sciences, Beijing 100700, China

**Keywords:** smoke-isolated karrikins, *Salvia miltiorrhiza*, tanshinone I, jasmonic acid, nitric oxide

## Abstract

Although smoke-isolated karrikins (KAR_1_) could regulate secondary metabolism in medicinal plants, the signal transduction mechanism has not been reported. This study highlights the influence of KAR_1_ on tanshinone I (T-I) production in *Salvia miltiorrhiza* and the involved signal molecules. Results showed KAR_1_-induced generation of nitric oxide (NO), jasmonic acid (JA) and T-I in *S. miltiorrhiza* hairy root. KAR_1_-induced increase of T-I was suppressed by NO-specific scavenger (cPTIO) and NOS inhibitors (PBITU); JA synthesis inhibitor (SHAM) and JA synthesis inhibitor (PrGall), which indicated that NO and JA play essential roles in KAR_1_-induced T-I. NO inhibitors inhibited KAR_1_-induced generation of NO and JA, suggesting NO was located upstream of JA signal pathway. NO-induced T-I production was inhibited by SHAM and PrGall, implying JA participated in transmitting signal NO to T-I accumulation. In other words, NO mediated the KAR_1_-induced T-I production through a JA-dependent signaling pathway. The results helped us understand the signal transduction mechanism involved in KAR_1_-induced T-I production and provided helpful information for the production of *S. miltiorrhiza* hairy root.

## 1. Introduction 

Smoke generating from burning plant material has been known to contain karrikins (KAR_1_)—chemicals that are powerful germination promoters. KAR_1_ plays a major role in natural systems as it is highly active at very low concentrations, shows great potential in agriculture [1] and is promising to be a new plant growth regulator [2,3,4,5,6,7,8]. Until now the effects and potential mechanisms of KAR_1_ on the accumulation of secondary metabolite in medicinal plants has not been reported.

*Salvia miltiorrhiza*, commonly known as ‘Danshen’ in Chinese, is one of the most renowned medicinal herbs in China. Its roots and rhizomes have been widely used to remove blood stasis and to eliminate carbuncle throughout Chinese history [9]. In recent years, Danshen has been widely used in medicine, food and cosmetics in European and American markets, which has increased the demand of *S. miltiorrhiza* [10]. The most important active constituents, tanshinones, are terpenoids. Terpenoids are the largest class of plant secondary metabolites. The biosynthesis of tanshinones can be traced to two distinct routes, the mevalonate pathway (MVA pathway) and the 2-C-methyl-D-erythritol-4-phosphate pathway (MEP pathway), in which a universal five-carbon isoprene precursor, ispentenyldiphosphate (IPP) is used as building block. Tanshinone-type constituents such as tanshinone I (T-I) are considered as major pharmacologically active components and important indexes for measuring the quality of Danshen [11,12]. Now, the supply of *S. miltiorrhiza* to the market mostly relies on field cultivation, so it is vital to take effective measures to improve the content of T-I in the cultivation. In our previous study, it has been found that treatments of plant-derived smoke-water (SW) could markedly increase the content of T-I in *S. miltiorrhiza.* However, we are not aware of the underlying mechanisms of KAR_1_ on the accumulation of T-I in *S. miltiorrhiza*.

The activation of endogenous signaling pathways has been well-documented to play key roles in regulating accumulation of secondary metabolites in plants [13,14]. Signaling molecules, such as nitric oxide (NO), jasmonic acid (JA) and the ‘cross-talk’ among them have gained great attention [15]. NO has emerged as a key signal role that exerts various signaling functions in the mechanism of multiple biological functions in plants [6,16,17,18,19]. JA plays an essential role in secondary metabolism in medicinal plants [20,21]. However, there is no information describing how these signaling molecules related to the KAR_1_-induced accumulation of tanshinones in *S. miltiorrhiza.* The hairy root culture system has been considered as a valuable tool for signal transduction research and a platform for mass production of bioactive components [11,12,22]. Biotic elicitors (yeast extracts), abiotic elicitors (silver ion, La) and plant signal material (JA) have been widely used in enhancing tanshinones production in *S. miltiorrhiza* hairy root [23,24,25]. JA increased the accumulation of tanshinone, about 5.8 times that of the control, and also up-regulated the expressions of most investigated genes in *S. miltiorrhiza* hairy root [26]. JA participated in yeast extracts-induced generation of tanshinones in *S. miltiorrhiza* [11]. This study aimed to investigate the roles of JA and NO and their ‘cross-talk’ in KAR_1_-caused generation of T-I in *S. miltiorrhiza*, which would help us preliminarily understand the mechanisms involved in KAR_1_-induced T-I production in *S. miltiorrhiza*.

## 2. Results and Discussion

### 2.1. KAR_1_-Induced Increasing of T-I in *S. miltiorrhiza* Hairy Root

*S. miltiorrhiza* hairy root was treated with and without KAR_1_ (control) to evaluate the influence of KAR_1_ on the generation of T-I. The effects of KAR_1_ on the content of T-I in *S. miltiorrhiza* were present in Figure 1. Treatment with KAR_1_ improved the content of T-I (205.13 mg/g) compared to the control (176.84 mg/g) at 24 h after KAR_1_ treatment. There is little literature on the influence of KAR_1_ on the production of secondary metabolite in medicinal plants. Aremu et al. [27] reported that treating *Tulbaghia ludwigiana* with smoke water caused a significant increase in the content of flavonoids compared to the control. Soós et al. [28] demonstrated that smoke water could upregulate the expression of genes and promote biosynthesis of phenolic compounds. Data obtained from this study indicated that KAR_1_ could enhance the content of T-I in *S. miltiorrhiza*, which implied that using KAR_1_ for enhancing the production of tanshinones has significant scientific and industrial implications in hairy root production.

### 2.2. Burst of NO and JA Induced by KAR_1_

The contents of NO and JA significantly fluctuated in *S. miltiorrhiza* treated with KAR_1_ compared to the control. It has not been found that the levels of NO and JA in the control show significant changes, indicating that the increase of NO and JA is not owing to development-dependent changes. As shown in Figure 2, NO content was improved significantly with treatment of KAR_1_, reaching 25.95% more than the control by 6 h (*p* < 0.05), 30.69% more by 12 h (*p* < 0.05) and 34.03% more by 48 h (*p* < 0.05) respectively. As displayed in Figure 3, JA levels in KAR_1_-pretreated hairy root displayed a time dependent increase, reaching the peak at 1.41-fold of control levels at 12 h after treatment (*p* < 0.05) and then decreased gradually but remained significantly higher (*p* < 0.05) than that of the control. A KAR_1_-caused burst of JA occurred later than generation of NO. It has been reported that NO and JA participated in the biosynthesis of matrine, and synergistic action of NO and JA in accumulation of matrine might be in virtue of the mutually amplifying reaction between NO and JA [29]. Previous studies have shown NO and Put (putrescine) are upstream signals that regulate ginsenoside synthesis during the adventitious roots culture of *Panax quinquefolius* [10].

NO played a pivotal role in the transcriptional regulation of genes related to the phenylpropanoid biosynthetic pathway in *Arabidopsis* and maize. It improved the expression of transcription factors encoding genes such as *ZmP*, *HY5* and *MYB12* and the content of flavonoid [22]. Ren and Dai [16] demonstrated NO-regulated external inducer-induced generation of volatile oil in *Atractylodes lancea*. It has been investigated that JA acted as a vital signal molecule that regulated secondary metabolism and defense response in plants. Xu et al. [30] identified an important induction effect of JA in heat-shock-induced sesquiterpene production in *Aquilaria sinensis*. Our results indicated that KAR_1_-induced generation of NO and JA occurred earlier than the accumulation of T-I. It is hypothesized that JA and NO may act as signal molecules in KAR_1_-induced generation of T-I in *S. miltiorrhiza*. Furthermore KAR_1_-induced NO generation occurred earlier than JA.

### 2.3. JA Acted as a Downstream Signal of NO Pathway Induced by KAR_1_

Although a burst of the two signal molecules suggests defensive reactions of the hairy root in response to KAR_1_, it is still uncertain about their possible upstream and downstream relationships. Thus, the influence of PBITU and cPITO on KAR_1_-caused JA generation and SHAM and PrGall on KAR_1_-induced NO have been investigated. Our test displayed that cPITO and PBITU significantly inhibited the burst of JA induced by KAR_1_ (*p* < 0.05, Figure 4), however, SHAM and PrGall have not been found to severely affect the generation of NO (Figure 5). It is not difficult to see that KAR_1_-induced NO generation located in upstream of JA biosynthesis.

### 2.4. Dependence of KAR_1_-Stimulated T-I Production on NO Accumulation as well as JA production

It has been exhibited in our experiments that NO generation and production of JA were early events in hairy root of *S. miltiorrhiza* responding to KAR_1_. Whilst little information about whether NO and JA participated in KAR_1_-induced accumulation of T-I has been known. So we investigated the influence of scavengers and inhibitors of JA and NO on production of T-I induced by KAR_1_. As displayed in Figure 6, cPTIO and PBITU significantly (*p* < 0.05) suppressed the increase of T-I induced by KAR_1_, suggesting that KAR_1_-induced accumulation of T-I through NO pathway. Treatments of SHAM and PrGall induced a decline in T-I level, indicating that JA plays a signal part in KAR_1_-induced increase of T-I. These results were verified by the finding that the suppression of inhibitors of JA and NO on increase of T-I induced by KAR_1_ were turned back by treatments of JAMe and SNP (Figure 4). Treatments of NO donor SNP significantly improved the content of T-I, which was evaluated as much as 92.70% of that of KAR_1_ response. SNP-stimulated increasing of T-I was significantly suppressed by SHAM and PrGall (*p* < 0.05). Production of T-I in *S. miltiorrhiza* was stimulated by treatment of JAMe, and it has not been inhibited by PBITU or cPITO. These results displayed that NO-triggered T-I generation depend on JA pathway. This conclusion was further supported by the finding that suppression of SHAM and PrGall on SNP-induced T-I production is relieved by treatment of JAMe (Figure 6).

## 3. Materials and Methods

### 3.1. Hairy Root Culture and Experimental Design

*S. miltiorrhiza* hairy root culture was established by infecting the leaf with *Agrobacterium rhizogenes* bacterium (ACCC10060). It was incubated in 6, 7-V medium, which contained sucrose of 30 g/L. Experiments in this study were carried out in 250-mL flasks on an orbital shaker running at 120 rpm and 25 °C in the dark [31].

After 18 days of culture, KAR_1_, signal molecular and scavengers were added into the medium and the samples were then allowed to continue culturing for additional days. The content of tanshinone I was determined until sampling for evaluation at adaptation time point. No elicitors were added to the control cultures. Chemical reagents used in the experiment were bought from Sigma Co. (St. Louis, MO, USA), including NO donor sodium nitroprusside (SNP), NO-specific scavenger 2-(4-Carboxyphenyl)-4,4,5,5-tetramethylimidazoline-1-oxyl-3-oxide (cPTIO), nitric oxide synthase (NOS) inhibitors, S,S′-1,3-phenylene-bis(1,2-ethanediyl)-bis-isothiourea (PBITU). Jasmonic acid methyl ester (JAMe), JA synthesis inhibitor salicylhydroxamic acid (SHAM) and JA synthesis inhibitor n-propylgallate (PrGall). Chemical reagent, which was dissolved in water or 0.2% dimethyl sulfoxide solution, was used in hair roots 36 h before treatments of KAR_1_-or signal molecules. Each treatment consisted of 10 replicates, and all treatments were repeated three times.

### 3.2. Preparation of KAR_1_ Solution

Smoke water was obtained with the method described by Light et al. [32]. Briefly, dry branches of *Crataegus pinnatifida* and *Magnolia denudata* were burned slowly with smoke but no flame, and the smoke was taken through 500 mL distilled water for 45 min. KAR_1_ was isolated and identified from smoke water with the method of Van Staden et al. [3] and 10^−9^ M was used in the experiment.

### 3.3. HPLC Analysis of T-I

The content of T-I in hairy root of *S. miltiorrhiza* was analyzed based on the methods of Liang et al. [33]. An oven-dried sample (0.2 g) was pulverized with a mortar and pestle, and extracted with 20 mL 70% methanol under ultrasonic treatment for 1 h. The resulting mixture was centrifuged at 8000 r/min for 20 min and filtered through a 0.22 μm syringe filters before high performance liquid chromatography (HPLC) analysis. The content of T-I was analyzed by HPLC on Agilent-1260 apparatus equipped (Palo Alto, CA, USA) with a C18 column (4.6 mm × 250 mm, 5 µm particle size), and the flow rate was 1 mL/min with the detection wavelength at 275 nm. The working temperature of column was kept at 30 °C and the sample injection volume was 20 μL. Separation was achieved by elution using a linear gradient with solvent-B (acetonitrile) and solvent-A (0.2%-methanoic acid-ammonium). The gradient was as follows: 0–20 min, 20–40% B; 20–21 min, 40–80% B; 21–40 min, 80–90% B; 40–45 min, 90–20% B.

### 3.4. Determination of NO

The content of NO was estimated in *S. miltiorrhiza* hairy root using the method of Li et al. [34] with slight modification. According to the principle of the conversion of oxyhemoglobin (HbO_2_) to methemoglobin (MetHb), the content of NO was determined by spectrophotometry (Shanghai Spectrum Instrument Co. Ltd., China) at 401 and 421 nm. NO accumulation in hairy root was labeled with a specific fluorescent probe of DAF-2DA (4-amino-5-methylamino-2′, 7′-difluorofluorescein diacetate).

### 3.5. Measurement of JA

The content of JA in *S. miltiorrhiza* hairy root was determined by the method described in the instruction manual of kit (Shanghai Enzyme Biotechnology Co., Ltd., Shanghai, China). Briefly, 4.0 mL of phosphate buffer was added to 1.0 g of hairy root; the mixture was uniformly ground in a mortar on an ice plate; and the homogenate was centrifuged at 2800 r/min for 20 min at 4 °C. The supernatant was obtained for the JA content assays. The absorbance was read at 450 nm.

### 3.6. Statistical analysis

ANOVA with SPSS software (version 18.0, SPSS, Inc., Chicago, IL, USA) was used analyze all data and statistical differences among treatments was based on one-way analysis of variance (ANOVA) and a significant difference was concluded at a level of *p* < 0.05.

## 4. Conclusions

In summary, the results from this work revealed KAR_1_ improved the production of T-I by triggering the biosynthesis of endogenous NO and JA in hairy root of *Salvia miltiorrhiza*. Furthermore, NO regulates the KAR_1_-induced T-I production through a JA-dependent signaling pathway. Together, the results suggest that KAR_1_ may be used as a new practical approach to improve the T-I accumulation in *S. miltiorrhiza* by modulating NO and JA levels. This information will help us better understand the underlying mechanism of KAR_1_-regulating secondary metabolism. Furthermore, it also suggests strategies to improve the quality of medicinal herbs. Whether these are other downstream molecules participating in JA signal transduction leading to increase of T-I in *Salvia miltiorrhiza* and their relationships with JA still remains unrevealed. Therefore, it is apparent that we are only at the early stage in understanding the signal transduction mechanism in *S. miltiorrhiza.* Moreover, this means that molecular biology would be used to provide molecular evidence for revealing the signal transduction mechanism in KAR_1_-regulated secondary metabolism in medicinal plants.

## Figures and Tables

**Figure 1 molecules-24-01229-f001:**
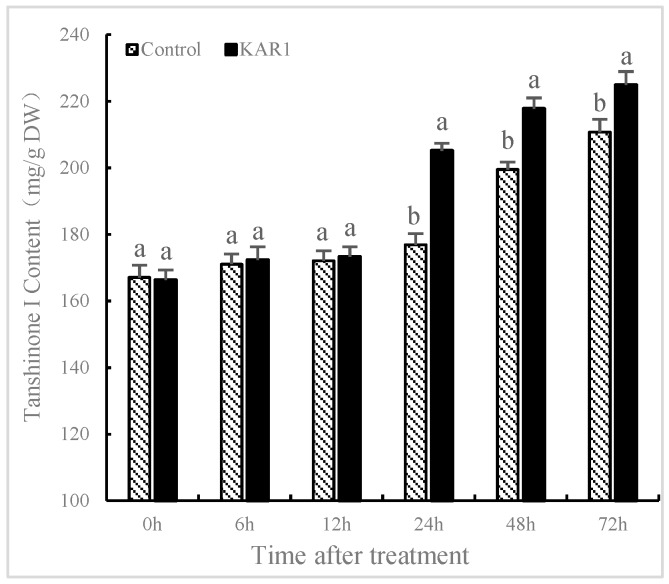
Effects of karrikins on the accumulation of tanshinone I in *S. miltiorrhiza* hairy root. Data are means of three replicates ± SD. Different letters indicate significantly different values according to one-way ANOVA followed by Tukey’s test (*p* < 0.05).

**Figure 2 molecules-24-01229-f002:**
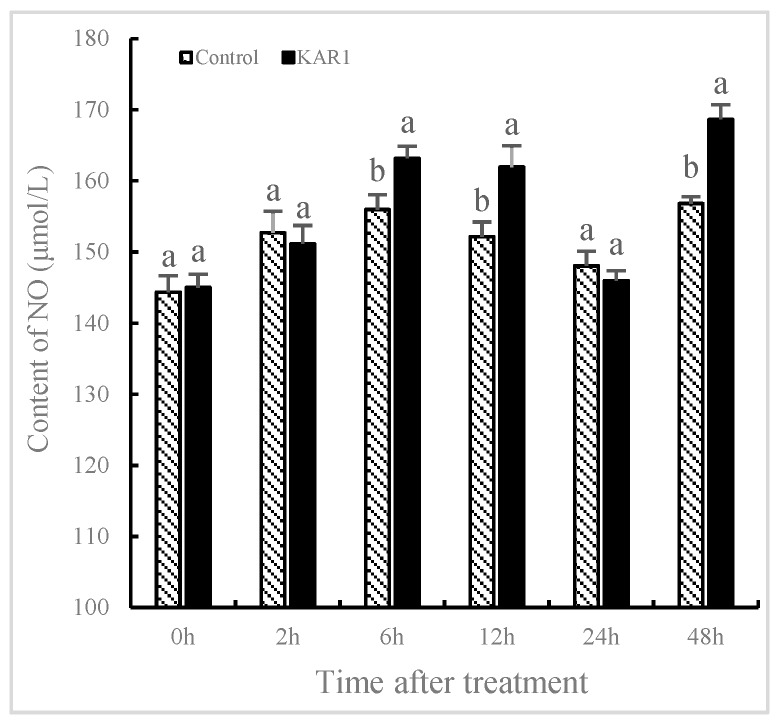
Time courses of NO level of *S. miltiorrhiza* hairy roots. The roots treated with KAR_1_ were harvested at determined time points. NO contents of the root were then determined. Data are means of three replicates ± SD. Different letters indicate significantly different values according to one-way ANOVA followed by Tukey’s test (*p* < 0.05).

**Figure 3 molecules-24-01229-f003:**
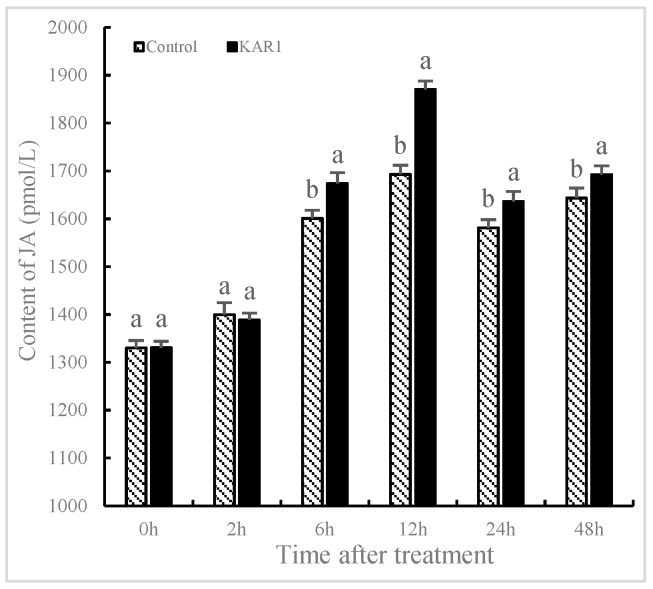
Time courses of jasmonic acid level of *S. miltiorrhiza* hairy root. The roots treated with KAR_1_ were harvested at determined time points. JA contents of the root were then determined. Data are means of three replicates ± SD. Different letters indicate significantly different values according to one-way ANOVA followed by Tukey’s test (*p* < 0.05).

**Figure 4 molecules-24-01229-f004:**
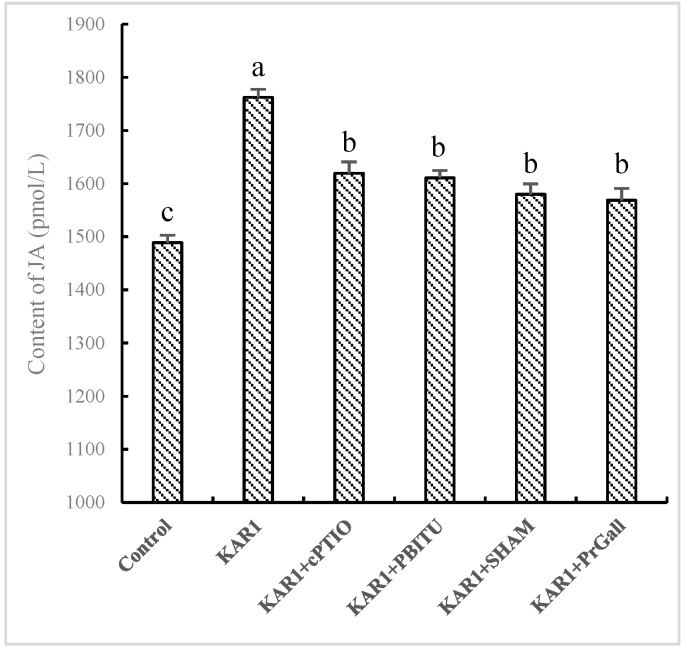
Effects of inhibitors on KAR_1_-induced JA accumulation in *S. miltiorrhiza* hairy root. *S. miltiorrhiza* hairy root treated with KAR_1_, and various inhibitors were harvested at 12 h after KAR_1_ and NO contents were determined. Inhibitors were pretreated 1 h before treatment of KAR_1_. The control received vehicle solvent only. Data are means of three replicates ± SD. Different letters indicate significantly different values according to one-way ANOVA followed by Tukey’s test (*p* < 0.05).

**Figure 5 molecules-24-01229-f005:**
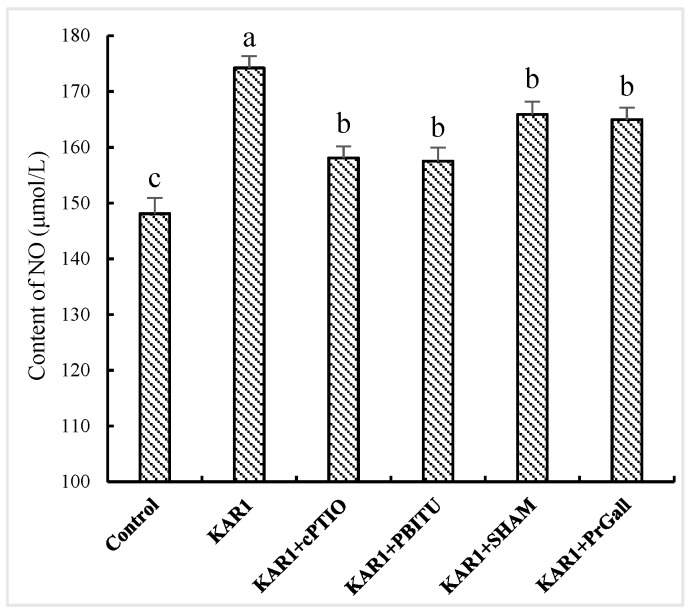
Effects of inhibitors on KAR_1_-induced NO generation in *S. miltiorrhiza* hairy root. *S. miltiorrhiza* hairy root treated with KAR_1_, and various inhibitors were harvested at 12 h after KAR_1_ and JA contents were determined. Inhibitors were pretreated 1 h before treatment of KAR_1_. The control received vehicle solvent only. Data are means of three replicates ± SD. Different letters indicate significantly different values according to one-way ANOVA followed by Tukey’s test (*p* < 0.05).

**Figure 6 molecules-24-01229-f006:**
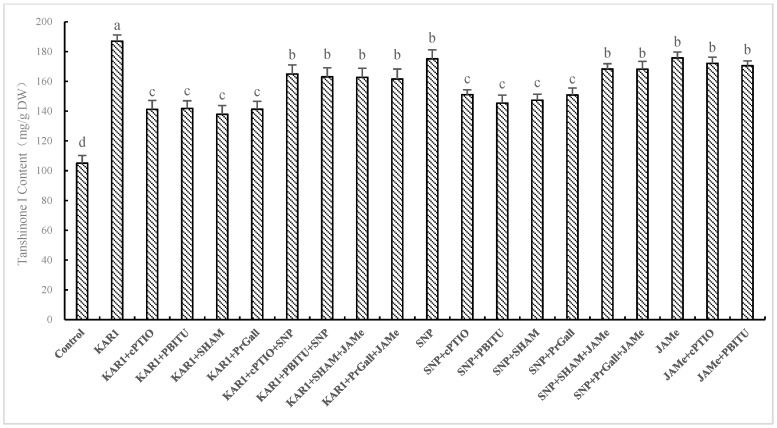
Effects of inhibitors on KAR_1_-induced T-I production of *S. miltiorrhiza* hairy root. The root treated with KAR_1_ of, and inhibitors were harvested at 24 h after KAR_1_ and T-I production was then determined. Inhibitors were pretreated 1 h before KAR_1_. The control received vehicle solvent only. Data are means of three replicates ± SD. Different letters indicate significantly different values according to one-way ANOVA followed by Tukey’s test (*p* < 0.05).

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
