# Peer review of "Smoke-Isolated Karrikins Stimulated Tanshinones Biosynthesis in Salvia miltiorrhiza through Endogenous Nitric Oxide and Jasmonic Acid"

_molecules, 2019, doi:10.3390/molecules24071229_

Round 1

Reviewer 1 Report

Overall the document requires extensive editing for language style and grammar.  

This manuscript describes an experiment done to determine the signaling pathways involved in the production of tanshinones(T-1) after smoke-karrikin (KAR1) application in hairy root cultures of S. miltiorrhiza. The authors describe a method for isolating and applying Kar1 from burnt plant material. They use a series of signaling cascade activators and inhibitors of NO and JA to do a functional analysis and to determine the up and down-stream components of the Kar1 to TI1 induction.  

The authors do not provide enough background information on the pathway for the production of tanshinones or their biological function or synthesis in plants. On line 41, they cite a study in which SW increased the content of T1 in plants, but do not define what SW is.   As Tanshinones are terpene components, perhaps they can provide a brief literature review on what is known about secondary metabolite diterpenoid production and what hormones and metabolites have been found to increase and decrease their production. 

There are significant problems with figure 1.  The authors claim throughout the manuscript (ie. Line 105) that "Our results indicated that KAR1-induced generation of NO and JA occurred earlier than the accumulation of T1.  However, in figure 1, T1 was sampled at 0, 24, 48 and 72 hours, while NO and JA were sampled at 2, 4, 6, 12, etc. hours.  Did the authors measure T1 production earlier than 24 hours and these data were just not reported?  This needs to be made clear in the manuscript otherwise, there is no data to support that T1 production occurs after NO and JA signaling.   Furthermore, with Figure 1, the authors do not provide any reasoning why the controls are statistically significantly different from the starting point controls. 

Additionally, I have never heard of using a colorometric method method for determining the concentration of JA in a sample.  Typically JA is measured using HPLC/MS and requires a specific C18 column.  The authors must cite additional studies that use this method or provide additional support for the efficacy of this method of quantifying JA. 

Figure 6 must be redrawn to be simpler. Samples showing NO vs. JA suppression must be grouped into categories.  As written, the image is hard to decipher.  I also suggest putting the names of the treatments directly on the columns so the reader does not have to switch back and forth from the caption to the graph. 

Overall, there are some significant flaws in the paper and methods. The results suggesting a Kar1--->NO---> JA---> T-1 pathway are overstated as using enzyme inhibitors is not the best deductive way to determine functionality.  Perhaps using knock-out mutants would be a better way to assess this. 

Author Response

Point 1: This manuscript describes an experiment done to determine the signaling pathways involved in the production of tanshinones (T-1) after smoke-karrikin (KAR1) application in hairy root cultures of S. miltiorrhiza. The authors describe a method for isolating and applying Kar1 from burnt plant material. They use a series of signaling cascade activators and inhibitors of NO and JA to do a functional analysis and to determine the up and down-stream components of the Kar1 to TI1 induction.

Point 2: The authors do not provide enough background information on the pathway for the production of tanshinones or their biological function or synthesis in plants. On line 41, they cite a study in which SW increased the content of T1 in plants, but do not define what SW is.   As Tanshinones are terpene components, perhaps they can provide a brief literature review on what is known about secondary metabolite diterpenoid production and what hormones and metabolites have been found to increase and decrease their production.

Response 2: The background information on the biosynthesis of tanshinones has been added in the introduction.

The following contentThe most important active constituents, tanshinones, are terpenoids. Terpenoids are the largest class of plant secondary metabolites. The biosynthesis of tanshinones can be traced to two distinct routes, the MVA pathway and the MEP pathway, in which a universal five-carbon isoprene precursor, ispentenyldiphosphate (IPP) is used as building block.has been added in Introduction part. 

On line 45, “SW” has been defined. 

On line 53, the following contentJA plays an essential role in secondary metabolism in medicinal plants [20-21].has been added in Introduction part on line 53.

On line 56, the following contentBiotic elicitors (yeast extracts), abiotic elicitors (silver ion, La) and plant signal material (JA) have been widely used in enhancing tanshinones production in S. miltiorrhiza hairy root [23-25]. JA increased the accumulation of tanshinone, about 5.8 times that of the control, and also up-regulated the expressions of most investigated genes in S. miltiorrhiza hairy root [26]. JA participated in yeast extracts-induced generation of tanshinones in S. miltiorrhiza [11].has been added in Introduction part.

Point 3: There are significant problems with figure 1.  The authors claim throughout the manuscript (ie. Line 105) that "Our results indicated that KAR1-induced generation of NO and JA occurred earlier than the accumulation of T1.  However, in figure 1, T1 was sampled at 0, 24, 48 and 72 hours, while NO and JA were sampled at 2, 4, 6, 12, etc. hours.  Did the authors measure T1 production earlier than 24 hours and these data were just not reported?  This needs to be made clear in the manuscript otherwise, there is no data to support that T1 production occurs after NO and JA signaling. Furthermore, with Figure 1, the authors do not provide any reasoning why the controls are statistically significantly different from the starting point controls.

Response 3: Treatment of KAR1 has not been observed to improve the content of T-I at 6 and 12 hours, so these data were not shown in the initial manuscript. Now these data have been added in Figure 1 for supporting that T-I production occurs after NO and JA signaling.

The tanshinones accumulates in Salvia miltiorrhiza over time, so the production of T-I in the control is significantly different from the starting point.

Point 4: Additionally, I have never heard of using a colorometric method method for determining the concentration of JA in a sample.  Typically JA is measured using HPLC/MS and requires a specific C18 column.  The authors must cite additional studies that use this method or provide additional support for the efficacy of this method of quantifying JA.

Response 4: Same to reviewers' comment 3.

Point 5: Figure 6 must be redrawn to be simpler. Samples showing NO vs. JA suppression must be grouped into categories.  As written, the image is hard to decipher.  I also suggest putting the names of the treatments directly on the columns so the reader does not have to switch back and forth from the caption to the graph.

Response 5: The names of the treatments have been directly putted on the columns in figure 4, figure 5 and figure 6.

Point 6: Overall, there are some significant flaws in the paper and methods. The results suggesting a Kar1--->NO---> JA---> T-1 pathway are overstated as using enzyme inhibitors is not the best deductive way to determine functionality.  Perhaps using knock-out mutants would be a better way to assess this.

Response 6: Thanks for reviewer’s good suggestions. Mutants would be a very good way to study signal pathways. However in recent years pharmacological strategy has been also considered as a good method to study signal pathways and ”cross-talk” among multiple signaling pathways in secondary metabolism in medicinal plants [1-7]. Pharmacological strategy has been used in our study to investigate the roles of JA and NO and their “cross-talk” in KAR1-caused generation of T-I in S. miltiorrhiza.

The following content has been added in conclusion. “Moreover means of molecular biology would be used to provide molecular evidence for revealing the signal transduction mechanism in KAR1-regulated secondary metabolism in medicinal plants.”

Reference:

1.      Wang, Y.J.; Shen, Y.; Shen, Z.; Zhao, L.; Ning, D.L.; Jiang, C.; Zhao, R.; Huang, L,Q. Comparative proteomic analysis of the response to silver ions and yeast extract in salvia miltiorrhiza hairy root cultures. Plant Physiol Biochem. 2016, 107, 364-373.

2.      Ren, C.G.; Dai, C.C. Nitric oxide and brassinosteroids mediated fungal endophyte-lnduced volatile oil production through protein phosphorylation pathways in Atractylodes lancea plantlets. J Integr Plant Biol. 2013, 55, 1136-1146.

3.      Wang, Y.; Dai, C.C.; Zhao, Y.W.; Peng, Y. Fungal endophyte-induced volatile oil accumulation in atractylodes lancea plantlets is mediated by nitric oxide, salicylic acid and hydrogen peroxide. Process Biochem. 2011, 46, 730-735.

4.      Xu, M.J.; Dong, J. F. Synergistic action between jasmonic acid and nitric oxide in inducing matrine accumulation of Sophora flavescens suspension cells. J Integr Plant Biol. 2008, 50, 92-101.

5.      Liu, Y.; Wang, R.; Zhang, P.; Chen, Q.; Luo, Q.; Zhu, Y.; Xu, J. The nitrification inhibitor methyl 3-4-hydroxyphenyl propionate modulates root development by interfering with auxin signaling via the NO/ROS pathway. Plant Physiol. 2016, 171, 1686.

6.      Xu, Y.H.; Liao, Y.C.; Zhang, Z.; Liu, J.; Sun, P.W.; Gao, Z.H.; Sui, C.; Wei, J.H. Jasmonic acid is a crucial signal transducer in heat shock induced sesquiterpene formation in Aquilaria sinensis. Sci. Rep. 2016, 6, 21843.

7.      Liu, W.; Li, R.J.; Han, T.T.; Cai, W.; Fu, Z.W.; Lu, Y.T. Salt stress reduces root meristem size by nitric oxide-mediated modulation of auxin accumulation and signaling in Arabidopsis. Plant Physiol. 2015, 168, 343–356.

Reviewer 2 Report

Authors aimed to reveal signal transduction mechanism involved in KAR1-induced T-I production in S. miltiorrhiza hairy root. Authors claimed KAR1 induced NO and JA in S. miltiorrhiza. Inhibitors for NO and JA were applied to figure out the relationship between T-I production and accumulation of NO and JA.

Is there any other plant signaling molecules involved in T-I production? Authors just measured NO and JA in hairy roots but other molecules. In order to better understand what molecules relate to KAR1 induced T-I production, other plant hormones should be measured in this study.

Author Response

Response to Reviewer 2 Comments

Point 1: Authors aimed to reveal signal transduction mechanism involved in KAR1-induced T-I production in S. miltiorrhiza hairy root. Authors claimed KAR1 induced NO and JA in S. miltiorrhiza. Inhibitors for NO and JA were applied to figure out the relationship between T-I production and accumulation of NO and JA.

Point 2: Is there any other plant signaling molecules involved in T-I production? Authors just measured NO and JA in hairy roots but other molecules. In order to better understand what molecules relate to KAR1 induced T-I production, other plant hormones should be measured in this study.

Response 2: Thanks for reviewer’s good suggestions.

The biosynthesis of secondary metabolic are regulated by multiple signal elements, such as salicylic acid, reactive oxygen species and et al. In our preliminary experiment it has been found that JA and NO may be involved in the accumulation of T-I. So in this study we highlighted the roles of JA and NO and their “cross-talk” in KAR1-caused generation of T-I in S. miltiorrhiza. Other signaling pathways will be further studied in the future.

The following content (Whether these are other downstream molecules participating in JA signal transduction leading to increase of T-I in Salvia miltiorrhiza and their relationships with JA still remain unrevealed. Therefore, it is apparent that we are only at the early stage in understanding the signal transduction mechanism in S. miltiorrhiza. Moreover means of molecular biology would be used to provide molecular evidence for revealing the signal transduction mechanism in KAR1-regulated secondary metabolism in medicinal plants) has been added in conclusion.

Round 2

Reviewer 1 Report

The authors have mostly addressed my concerns with the manuscript, especially adding to the introduction and re-labeling the graphs.  This definitely improves the overall flow and clarity of the paper. 

The column labels/descriptions in the captions for figures 5-8 mentioning A, B, C, D. etc can be deleted as the columns are no longer labeled that way. 

The authors, however, have still not addressed my concerns with their method for measuring JA.  As mentioned previously, JA has been measured in thousands of papers and always has been done using spectroscopy.  I have never read about any method that measures JA colorimetrically, which admittedly, would be a vastly simpler protocol than HPLC/MS.  As a reviewer, I am very hesitant to approve any paper where cross-talk with different hormones is being assessed when the mode for measuring those hormones is undocumented elsewhere in the literature.  I highly encourage the authors to cite other papers that use this JA method or provide proof of it's efficacy compared to HPLC/MS. 

Author Response

1.         The authors have mostly addressed my concerns with the manuscript, especially adding to the introduction and re-labeling the graphs.  This definitely improves the overall flow and clarity of the paper.

2.         The column labels/descriptions in the captions for figures 5-8 mentioning A, B, C, D. etc can be deleted as the columns are no longer labeled that way.

Response: “A, B, C, D….”has been deleted.

3.         The authors, however, have still not addressed my concerns with their method for measuring JA.  As mentioned previously, JA has been measured in thousands of papers and always has been done using spectroscopy.  I have never read about any method that measures JA colorimetrically, which admittedly, would be a vastly simpler protocol than HPLC/MS.  As a reviewer, I am very hesitant to approve any paper where cross-talk with different hormones is being assessed when the mode for measuring those hormones is undocumented elsewhere in the literature.  I highly encourage the authors to cite other papers that use this JA method or provide proof of it's efficacy compared to HPLC/MS.

Response: Thanks for reviewer’s good suggestions. The method of HPLC-MS is a good method to detect JA in plants. However, the enzyme-linked immunosorbent assay (ELISA) used in our manuscript has also been used for the determination of JA in plants [1-4]. Reference:

Ref. 1 Deng, A.X.; Tan, W.M.; He, S.P.; Liu, W.; Nan, T.G.; Li, Z.H.; Wang, B.M.; Li, Q.X. Monoclonal antibody-based enzyme linked immunosorbent assay for the analysis of jasmonates in plants. J Integr Plant Biol 2008, 50, 1046-1052.

Ref. 2 Zhao, J.; Li, G.; Yi, G.X.; Wang, B.M.; Deng, A.X.; Nan, T.G.; Li, Z.H.; Li, Q.X. Comparison between conventional indirect competitive enzyme-linked immunosorbent assay (icELISA) and simplified icELISA for small molecules. Anal Chim Acta 2006, 571, 79-85.

Ref. 3 Gan, L.J.; Xia, K.; Zhou, X. Fluctuations of jasmonates in all floret parts during anthesis process in wheat. Journal of Nanjing Agricultural University 2005, 28, 26-29.

Ref. 4 Wang, S.C.; Xu, L.L.; Li, G.J.; Chen, P.Y.; Xia, K.; Zhou, X. An ELISA for the determination of salicylic acid in plants using a monoclonal antibody. Plant Sci 2002, 162(4), 529-535.

In the future, we will compare the efficacy of the two methods.

Thanks for your good suggestions again!

Reviewer 2 Report

It will be better if authors could add response to question 1 to main manuscript.

Author Response

1.       It will be better if authors could add response to question 1 to main manuscript

Response: Related content has been added to the manuscript.